# Regulating photosalient behavior in dynamic metal-organic crystals

Samim Khan [1], Basudeb Dutta[1], Sanobar Naaz[1], Aditya Choudhury[2], Pierre-Andre Cazade[3,4], Emma Kiely[3], Sarah Guerin [3,4✉], Raghavender Medishetty [2✉] & Mohammad Hedayetullah Mir [1✉]

Dynamic photoactuating crystals have become a sensation due to their potential applications in developing smart medical devices, molecular machines, artificial muscles, flexible electronics actuators, probes and microrobots. Here we report the synthesis of two iso-structural metal-organic crystals, [Zn(4-ohbz)$_2$(4-nvp)$_2$] (**1**) and [Cd(4-ohbz)$_2$(4-nvp)$_2$] (**2**) {H4-ohbz = 4-hydroxy benzoic acid; 4-nvp = 4-(1-naphthylvinyl)pyridine} which undergo topochemical [2 + 2] cycloaddition under UV irradiation as well as sunlight to generate a dimerized product of discrete metal-complex [Zn(4-ohbz)$_2$(*rctt*-4-pncb)] {*rctt*-4-pncb = 1,3-bis(4'-pyridyl)-2,4-bis(naphthyl)cyclobutane} (**1'**) and one-dimensional coordination polymer (1D CP) [Cd(4-ohbz)$_2$(*rctt*-4-pncb)] (**2'**) respectively, in a single-crystal-to-single-crystal (SCSC) process. The Zn-based compound demonstrates photosalient behaviour, wherein crystals show jumping, splitting, rolling, and swelling upon UV irradiation. However, the Cd-based crystals do not show such behaviour maintaining the initial supramolecular packing and space group. Thus the photomechanical behaviour can be induced by choosing a suitable metal ion. The above findings are thoroughly validated by quantitative density functional theory (DFT) calculations which show that the Zn-based crystal shifts towards an orthorhombic structure to resolve the anisotropic UV-induced mechanical strain. Furthermore, the mechano-structure-property relationship has been established by complimentary nanoindentation measurements, which are in-line with the DFT-predicted single crystal values.

[1] Department of Chemistry, Aliah University, New Town, Kolkata 700156, India. [2] Department of Chemistry, IIT Bhilai, Sejbahar, Raipur, Chhattisgarh 492015, India. [3] Department of Physics, Bernal Institute, University of Limerick, Limerick V94 T9PX, Ireland. [4] SSPC, The Science Foundation Ireland Research Centre for Pharmaceuticals, University of Limerick, Limerick V94 T9PX, Ireland. ✉email: Sarah.Guerin@ul.ie; raghavender@iitbhilai.ac.in; chmmir@gmail.com

Evolution occurs due to the phenomenon known as natural selection, where organisms adapt and change to survive in their environment. Natural selection is based on the idea of "survival of the fittest", where the most adaptive organisms will gradually change or evolve as the environmental changes via locomotion[1]. Animal locomotion is seemingly autonomous, e.g. walking, running, swimming, jumping, hopping, flying, soaring and gliding. In the case of plants, a variety of mechanisms are employed in order to achieve their fast movements, e.g. the fast closing trap (100 ms) of the venus flytrap[2] or the opening of petals of the dogwood bunchberry's flower (0.5 ms). Some plants are able to move their leaves very rapidly in response to mechanical stimuli[3], with many plants spreading their seeds or pollen by rapid movement. Cardamine hirsuta has seed pods which explode on touching. Some beans twist as they dry out, putting tension on the seam, which at some point will split suddenly and violently flying the seeds meters from the maternal plant[4].

These locomotions can be mimicked and utilized in molecular crystals at the macroscopic level[5]. As Mexican beans respond to heat, small crystals can also be made respond to external stimuli such as heat (thermosalient)[6] or light (photosalient)[7] by jumping, swelling, or bursting. In this regard, a number of organic and metal-organic crystals have been found to naturally exhibit bending, twisting and swelling upon light illumination[8–11]. Beyond this, some crystals may burst, scatter, jump, hop, curl, coil, swim, and at the extreme level fragment into pieces– commonly known as the "photosalient" (PS) effect[12–17]. Underlying the concept of such photomechanical behavior is the accumulation of stress due to anisotropic expansion/contraction of unit cell volume via photoinduced structural changes, which eventually releases in the form of macroscopic mechanical motion of crystals[10,11]. Out of various PS effects, structural change via photochemical [2 + 2] cycloaddition is one of the most commonly utilized chemical tools in the study of such phenomena[18]. For thorough insight into organic synthesis using [2 + 2] processes in the solid state readers are directed to the foundational works of MacGillivray[19].

Among various photomechanical phenomena, the PS effect involves single crystals that violently explode and shatter into pieces[20,21]. In terms of actuation; this is a single time event due to the disintegration of the crystals[22,23]. Furthermore, there lies a gray area between the PS effect and single-crystal-to-single-crystal (SCSC) transformation. While SCSC transformations provide exact structural insights into the transformed structures[18,24–26], it is elusive in the case of PS effects as the single crystals are normally broken into small pieces[18]. Predominantly the structural changes are analyzed from a partially dimerized structure or from the recrystallized product[27–29]. However, the photoreaction may not be fully realised in the whole crystal due to a high absorbance of molecules in the crystal or potential side reactions. Achieving a clean SCSC transformation at the end of a violent PS reaction is a challenge. Herein, photocrystallography[30–32] is an important concept that brings us closer to being able to watch solid-state processes occur in real time (either in a metastable or short-lived excited state) during the determination of a single crystal of a complex and thus we may have better insight into the photo-conversion process of crystals.

Here, we report two iso-structural photoreactive metal-organic crystals of formulae [Zn(4-ohbz)$_2$(4-nvp)$_2$] (**1**) and [Cd(4-ohbz)$_2$(4-nvp)$_2$] (**2**) {H4-ohbz = 4-hydroxy benzoic acid; 4-nvp = 4-(1-naphthylvinyl)pyridine} that undergo topochemical [2 + 2] cycloaddition under UV light as well as sunlight to generate a dimerized product of a discrete metal-complex [Zn(4-ohbz)$_2$(*rctt*-4-pncb)]{*rctt*-4-pncb = 1,3-bis(4′-pyridyl)-2,4-bis(naphthyl)cyclobutane} (**1′**) and one-dimensional coordination

polymer (1D CP) [Cd(4-ohbz)$_2$(*rctt*-4-pncb)] (**2′**) via a SCSC process respectively. Interestingly, during this photoreaction, the Zn-complex **1** shows mechanical motion such as swelling, splitting, jumping and scattering. However, after the photomechanical effect is induced, these PS crystals uniquely maintain their single crystallinity nature and allow for single crystal structural elucidation. This is a rare example of a metal-complex that exhibits a single crystal structure even after PS effect. The Cd(II)-based crystal **2** does not demonstrate this PS effect although it maintains its single crystal nature. In addition, we have obtained thick crystals of [Zn(4-ohbz)$_2$(4-nvp)$_2$] (**p1**) by maintaining the reaction mixture for a long time, which exists as a supramolecular isomer of **1**. In **p1** crystals, the 4-nvp ligands are not aligned and thus are found to be photoinert. Various experimental evidence has been previously sought to establish a structure-property relationship for the PS effect in crystals. Meanwhile, the PS effect of metal-complex is seldom explained by experimental and theoretical predictions. Here, we predominantly use density functional theory (DFT) to predict the mechanical properties of each crystal to understand the atomic-scale mechanisms and mechanical shifts that occur under irradiation. Outside of our previous work on high-accuracy screening of mechanical properties and phenomena, herein DFT calculations serve to understand the relative internal stress build-up in an irradiated photosalient crystal. Nanoindentation measurements, a reliable technique to check the mechano-structure-property relationship of the crystals, have been performed and the results are correlated to DFT-predicted values. In line with recent studies using single crystal DFT predictions, the results obtained from nanoindentation studies are well corroborated with calculated Young's Modulus values.

## Results and discussion

**Synthesis and characterization of 1, 2 and p1**. The compound [Zn(4-ohbz)$_2$(4-nvp)$_2$] (**1**) is synthesized by a slow diffusion method via layering of H4-ohbz and 4-nvp ligands onto a solution of Zn(NO$_3$)$_2$·4H$_2$O in the presence of Et$_3$N. Single crystal X-ray diffraction (SCXRD) analysis reveals that compound **1** crystallizes in the monoclinic space group $P2_1/c$ with Z = 8. In the solid-state structure, a pair of 4-nvp ligands from two adjacent moieties produce a head-to-tail arrangement with a distance between the centers of two C = C bonds of 3.79 Å (Fig. 1a), which is highly favorable for dimerization according to Schmidt's criteria (<4.2 Å)[33]. [Cd(4-ohbz)$_2$(4-nvp)$_2$] (**2**) is also synthesized by a similar procedure except using Cd(NO$_3$)$_2$·6H$_2$O in place of Zn(NO$_3$)$_2$·6H$_2$O. The compound **2** is isostructural to **1** and crystallizes in the monoclinic space group $P2_1/c$ with Z = 8. Similar to compound **1**, in the solid-state structure of **2**, a pair of 4-nvp ligands from two adjacent moieties produces a head-to-tail arrangement with a distance between the centers of two C = C bonds of 3.77 Å, which is also congenial for photodimerization. In addition, by keeping the reaction mixture of **1** for a long time, thick crystals of [Zn(4-ohbz)$_2$(4-nvp)$_2$] (**p1**) have been obtained, in which the 4-nvp ligands are not aligned and thus the crystals are expected to be photoinert.

**[2 + 2] Photocycloaddition reactions**. We performed the photoirradiation of **1** for 5 min under UV light (λ ~ 350 nm) wherein crystals have broken into fragments. The photoirradiation of light yellow single crystals of **1** yielded yellow-colored fragmented crystals. Then, we collected SCXRD data of the irradiated single crystal which reveals the formation of **i1** with partially dimerized cyclobutane rings derived from the olefinic bonds of 4-nvp in an SCSC manner (Fig. 1b). We have also taken SCXRD data of the

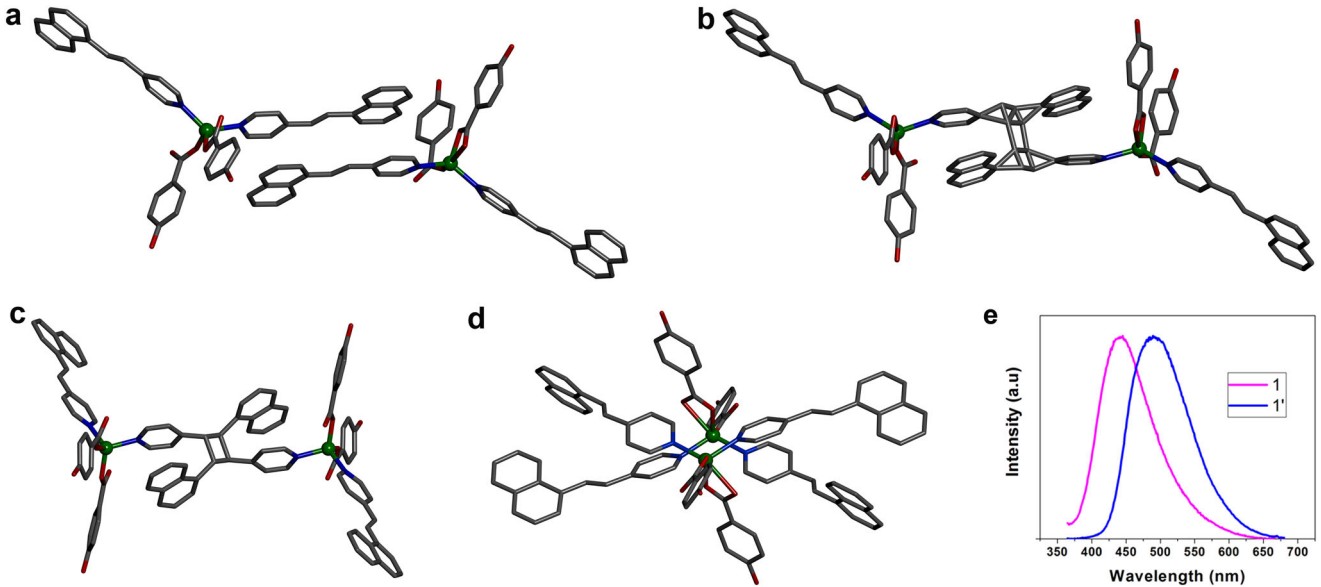

**Fig. 1 Crystal structure of the synthesized compounds. a** A perspective view of the compound **1**. **b** Intermediate compound **i1** via partial photdimerization. **c** dimerized compound **1′**. **d** Supramolecular isomer **p1**. Hydrogen atoms are not shown for clarity. **e** Solid-state photoluminescence of **1** and **1′** (Excitation wavelength 350 nm with slit 3 nm).

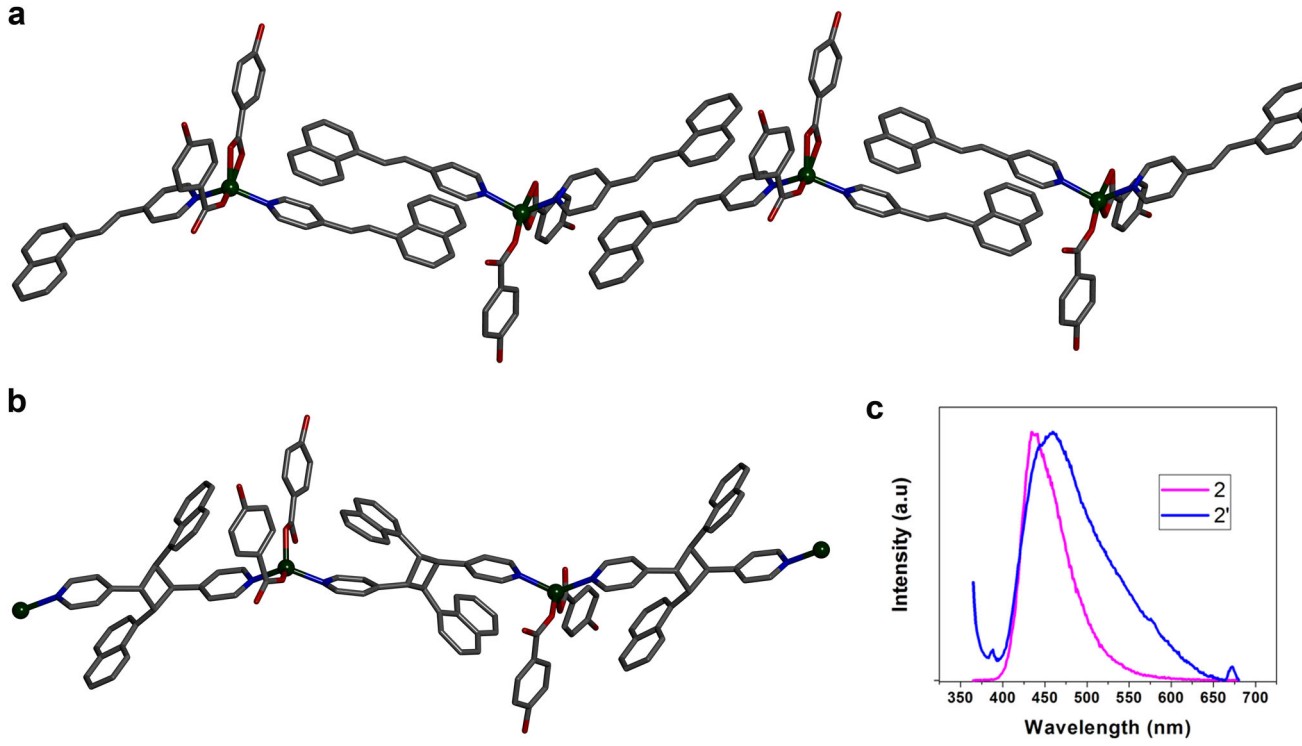

**Fig. 2 Crystal structure of the synthesized compounds. a** A perspective view of the compound **2**. **b** 1D CP **2′** formed via [2 + 2] photodimerization. Hydrogen atoms are not shown for clarity. **c** Solid-state photoluminescence of **2** and **2′** (Excitation wavelength 350 nm with slit 3 nm).

single crystal irradiated for 30 min which shows the 100% dimerized cyclobutane ring of compound **1′** (Fig. 1c). Meanwhile, a supramolecular isomer of **1**, the crystal **p1** does not undergo photodimerization as the 4-nvp ligands are not aligned to meet the criteria governed by Schmidt (Fig. 1d)[33]. Again, we kept the single crystal of **2** in the sunlight for 3 min and immediately collected the SCXRD data which reveals the formation of **2′** with fully dimerized cyclobutane rings derived from the olefinic bonds of 4-nvp in an SCSC manner (Fig. 2).

**Understanding photoreactivity and photosalient effects**. The photoreactivity of crystals **1** and **2** was investigated by [1]H NMR spectroscopy. The [1]H NMR spectra of **1′** shows the emergence of the cyclobutane peaks at ca. 5.2 ppm and the disappearance of olefinic double bonds at ca. 8.35 and 7.30 ppm along with the shifting of other protons (Supplementary Fig. 1–3). Similar observations have also been realized in case of compound **2′** (Supplementary Fig. 4, 5). In order to confirm the [1]H NMR results (Supplementary Figs. 2–5) we attempted to obtain

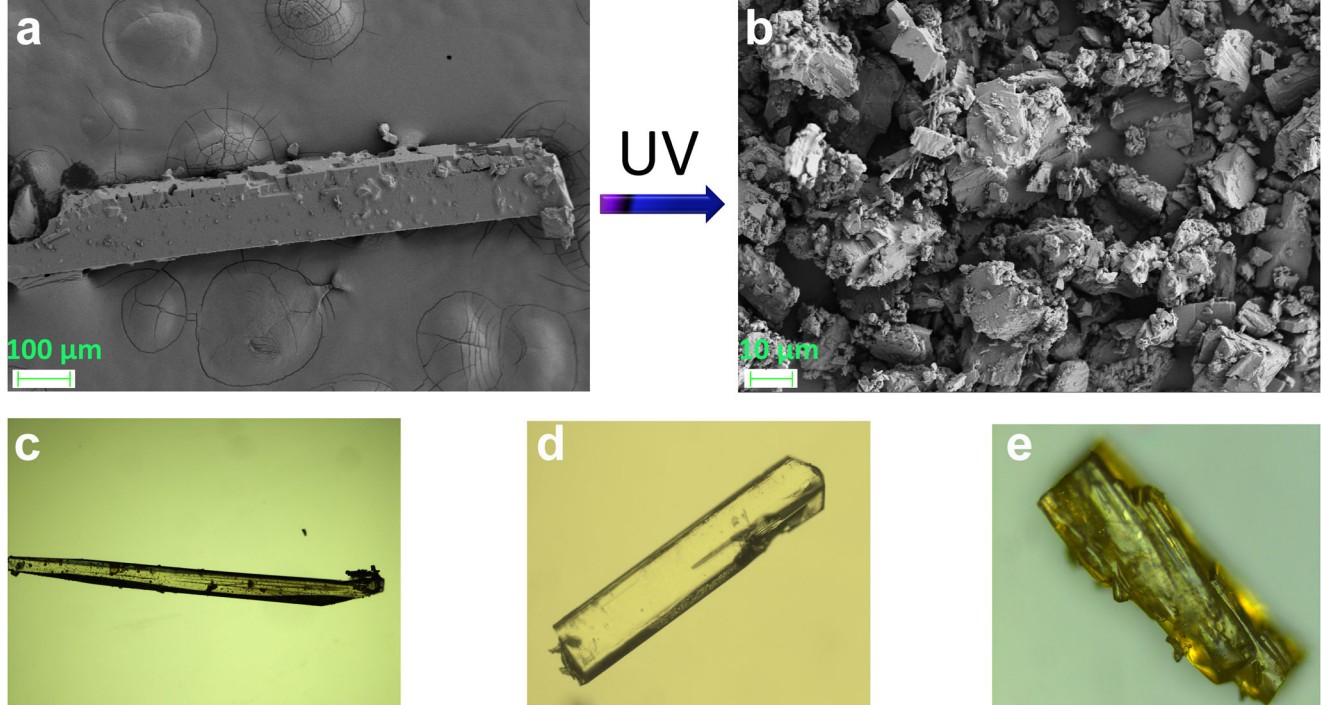

**Fig. 3 Illustration of PS effect. a** FESEM images of crystal **1** before UV irradiation. **b** FESEM images of crystal **1** after popping under UV irradiation. **c** Optical microscopic image of needle crystal **1**. **d** Optical microscopic image of block-shaped crystal **1**. **e** Optical microscopic image of plate-like crystal **p1**.

SCXRD data of dimerized compound **1** as SCXRD can only give the structure of the photoproduct unequivocally. In this aspect, we irradiated the single crystals under a microscope equipped with a high resolution camera in the presence of UV light. Interestingly we observed numerous photomechanical effects including violent jumping, splitting and scattering into pieces (Supplementary Movies 1–3). The PS effect can also be realized by field emission scanning electron microscopy (FESEM) images of the crystals before and after photoirradiation (Fig. 3a, b). However, unusually a piece of single crystal obtained after UV irradiation was found suitable for SCXRD data collection which revealed the formation of cyclobutane product **1′**. It should be mentioned that determining the structure of a PS crystal is extremely difficult as in most cases we obtain only debris of any crystal after PS effect. The only way to obtain the structure is either by recrystallization of the photoproduct or SCSC transformation, which is also extremely difficult to obtain. To the best of our knowledge, there is only one report published by Vittal et al., where the structure of the dimerized product of a PS crystal triggered by photochemical [2 + 2] cycloaddition could be determined by SCSC transformation[18]. Apart from this, we are not aware of any PS crystals based on metal-complexes that maintain single crystallinity. On the other hand, compound **2** does not show such PS effect although we have been able to obtain the crystal structure of photoproduct **2′**. Further, it should be mentioned that the crystals with long needle morphology (Fig. 3c) shows fast popping and break into pieces immediately after irradiation (Supplementary Movie 1). However, thick block-shaped crystals (Fig. 3d) show slow mechanical effect and undergo cracking after a long exposure of UV light (Supplementary Movie 4). In contrast, the **p1** crystal (Fig. 3e) does not show such a photomechanical effect even after a long period of irradiation (Supplementary Movie 5).

In order to deconstruct the PS effect in **1** and not **2**, we looked at the changes in cell parameters before and after photoirradiation. In case of **1**, there are changes in cell axes as well as cell volume after dimerization; however, compound **2** remains almost same in terms of cell parameters (Supplementary Table 1). The cell parameters reveal that the cell volume is also decreased by 227 $Å^3$ (3%) from **1** to **1′**, compared to 150 $Å^3$ (2%) from **2** to **2′**. It is well documented that the PS effect, as triggered by photodimerization is observed when there are substantive amount of changes in unit cell parameters. In that case, if the cell volume is decreased or increased during photodimerization (nanoscale property) then the crystals cannot withstand the extra stress and release this strain as mechanical motion (macroscopic property). In addition, if we take a closer look at the structure of the dimerized products, we observe that compound **1** undergoes 50% photodimerization and yields a discrete dimeric metal-complex, because one out of two aligned 4-nvp ligands (per metal center) moves away during photocycloaddition (Supplementary Fig. 1). This allows movement of the molecules and thus accumulated strain is released as PS effect of crystals. However, unlike **1**, compound **2** undergoes quantitative (100%) dimerization leading to the formation of a 1D chain polymer (Fig. 2), which tends to restrict the movement of the molecules. Besides the major peaks of PXRD plot of as-synthesized crystalline powders match well with those simulated from single crystal data which clearly indicates the phase purity of bulk materials before and after UV irradiation (Supplementary Figs. 6–9).

**DFT calculated mechanical properties**. In order to investigate the anisotropic mechanical properties of PS crystals, DFT calculations were carried out to predict the elastic stiffness tensor of each crystal, from which the calculated Bulk, Shear, and Young's Moduli can be derived[34]. The elastic stiffness tensor represents the anisotropic stress strain response of a material[35], where a tensor component $c_{ij}$ represents the stiffness of a material strained in direction i with an induced stress in direction j[36]. The numbers 1, 2 and 3 correspond to longitudinal responses in Cartesian directions x, y, and z (and also crystallographic axes a, b, and c in cubic or orthorhombic symmetries), and the numbers 4, 5, and 6

correspond to shear stress/strain responses along those same respective axes. Our calculations predict the full thirty-six component tensor for each crystal, which is shown in supporting information. For clarity, only the diagonal elements are shown and discussed in the main text i.e., the stress-strain response along each longitudinal and shear axis. Using the software ELATE the elastic moduli have been derived from this tensor, namely:

The Young's Modulus is by definition a measure of the ability of a crystal to withstand changes in length when under lengthwise tension or compression. It can be thought of as a bulk average of the anisotropic elastic stiffness tensor components, taking into account the dimensions of the system if calculated using the methods of Nye.

The Bulk Modulus of a substance is a measure of how resistant to compression the substance is. It is defined as the ratio of the infinitesimal pressure increase to the resulting relative decrease of the volume i.e., how the volume of the crystal responds to a force acting equally on all faces of the crystal.

The Shear Modulus is a measure of a crystal's resistance or response to shear stress. Similarly to the Young's Modulus the Shear Modulus can be conceptualised of as a bulk average of the shear elastic stiffness tensor contributions, though the mathematics is more complex than this. These moduli are presented as averages of the widely-used Voigt, Reuss, and Hill approximations[37,38], with the standard deviation being across these three approximations. Our predictions of elastic stiffness tensors have been extensively benchmarked across a number of publications on functional crystals[39–41], and experimental mechanical characterisation techniques[34], allowing for strong confidence in their accuracy and predictability (Supplementary Fig. 10). The moduli of crystals **1** and **2** are very similar (Table 1), with predicted Young's Modulus values of 9 GPa. Crystal **1** has slightly higher Bulk and Shear Modulus values of 11.1 and 3.3 GPa, versus average values of 10.6 and 2.8 GPa for crystal **2**. Delving deeper into the anisotropy we see more commonalities between the two crystals- in particular low $c_{66}$ values of 0.7 and 0.5 GPa for crystal **1** and **2** respectively. It is likely that the low shear stiffness along the crystallographic $a$ and $c$ axes facilitates the dimerization observed during UV irradiation. This trend in shear flexibility, along with stability along the perpendicular $b$ axis ($c_{55} = 7.5$ and 7.8 GPa), has been observed in our previous work on organic salts that demonstrate functional mechanical behaviors[40]. Crystal **2**, which has not demonstrated photosalience in this study shows less mechanical anisotropy and more flexibility than crystal **1**.

By obtaining a high-resolution XRD structure of the crystals as they are undergoing photosalience, we can use DFT to obtain dynamic information about how this phenomenon evolves over time, even with thermodynamically stable crystal structures. To understand what happens mechanically during the process of photosalience we can look at the trend in elastic stiffness going from crystal **1** to **i1** to **1′** (Fig. 4). After 5 min of irradiation, crystal **i1** is predicted to be stiffer than crystal **1**. The UV light

seemingly strengthening some intermolecular interactions (i.e., during cycloaddition) resulting in an increased Young's Modulus of 10.0 GPa. The low shear stiffness along the $c$ axis disappears, with $c_{66}$ more than doubling to 2.0 GPa and matching the predicted stiffness along the $a$ axis. However, the bulk modulus is predicted to decrease slightly to 10.7 GPa due to decreases in the $c_{11}$, $c_{44}$ and $c_{55}$ values. This quantitatively validates that irradiation causes mechanical restriction in some directions, while facilitating movement in others- a good way to induce spontaneous breakage or movement.

Finally looking at crystal **1′** we can observe the mechanical properties after popping/breaking. Crystal **1′** is predicted to be by far the most mechanically stable crystal, with the lowest mechanical anisotropy and highest Bulk Modulus of 12.0 GPa. While the longitudinal stiffness shows very little change compared to both **1** and **i1**, the shear stiffness has changed dramatically. The $c_{66}$ value has continued to increase, more than doubling again to a predicted value of 4.6 GPa. Here, $c_{44}$ increases to a near identical value of 4.7 GPa. The $c_{55}$ value has plummeted to 1.3 GPa, causing the Shear Modulus to decrease to 2.9 GPa. This is a contrast to crystal **2′** i.e. the non-photosalient crystal after irradiation. Crystal **2′** shows no change in $c_{44}$ or $c_{55}$ as a result of irradiation, with a small increase in $c_{66}$ from 0.5 to 0.9 GPa. Crystal **2′** has the lowest shear modulus of all the crystals in this study, with a predicted value of 2.1 GPa. Looking at the other negative control, the crystal **p1**, we again see that this crystal has already achieved mechanical stability, with the highest predicted Young's Modulus value of 11.2 GPa, and almost identical shear stiffness values to the post-irradiation crystal **1′**. Full DFT-calculated elastic tensors for crystals **1** and **2** are depicted in Supplementary Table 8.

Figure 5 shows the crystal structures of crystals **1**, **i1**, **1′**, and **2**. To deconstruct the difference in mechanical properties at the atomic scale is non-trivial, as the crystals are structurally very similar. The key difference is a shift from monoclinic to orthorhombic symmetry during irradiance. Crystal **1** has a monoclinic angle of 103.11°, which expands to 103.61° under irradiance (structure **i1**). After irradiance, this has changed dramatically to 90.98°. This shift towards higher symmetry changes the range of motion along each crystallographic plane, and thus changes the shear stiffness and increases bulk mechanical stability as discussed previously. The increase in atomic radius of the cadmium ions in crystal **2** result in a minor increase in monoclinic angle of 103.14°, with other negligible change in lattice parameters and volume. The change in mechanical properties during irradiation is a direct result of the aforementioned changes in unit cell parameters. From a supramolecular packing perspective, photosalience facilitates a significant rearrangement of the 4-ohbz chains. The increased volume and change in symmetry observed in crystal **1′** allows these molecular moieties to relax and stretch out (Fig. 5c), preferring this to the constrained 'wrinkled' arrangement of

**Table 1 DFT-calculated elastic stiffness tensor values and derived moduli (all values in GPa).**

| Elastic stiffness constant (GPa) | 1 | i1 | 1′ | p1 | 2 | 2′ |
|---|---|---|---|---|---|---|
| $c_{11}$ | 17.6 | 15.7 | 15.6 | 15.8 | 14.6 | 13.9 |
| $c_{22}$ | 23.2 | 23.3 | 20.7 | 12.7 | 17.4 | 14.8 |
| $c_{33}$ | 16.8 | 17.4 | 18.9 | 21.4 | 18.8 | 11.1 |
| $c_{44}$ | 2.8 | 2.1 | 4.7 | 4.9 | 7.5 | 7.5 |
| $c_{55}$ | 7.8 | 7.2 | 1.3 | 1.8 | 2.9 | 2.3 |
| $c_{66}$ | 0.7 | 2.0 | 4.6 | 3.8 | 0.5 | 0.9 |
| Young's Modulus | 9.0 ± 2.7 | 10.0 ± 1.7 | 8.1 ± 1.8 | 11.2 ± 0.4 | 9.0 ± 3.0 | 10.1 ± 0.9 |
| Bulk Modulus | 11.1 ± 0.1 | 10.7 ± 0.2 | 12.0 ± 0.7 | 9.0 ± 0.7 | 10.6 ± 0.4 | 5.7 ± 3.7 |
| Shear Modulus | 3.3 ± 1.1 | 3.7 ± 0.7 | 2.9 ± 0.7 | 3.2 ± 0.3 | 2.8 ± 1.2 | 2.1 ± 1.4 |

crystal **1** and **i1**. These calculations provide a molecular mechanism for the hypothesis that photosalience is a "macroscopic manifestation of the stress that can develop in response to the mechanical force created inside a crystal as a result of miniscule structure perturbation"[42,43]. Future calculations will expand on this work to look at dynamical stress-strain responses across different ligand-metal combinations, as well as examining volume expansion effects[44,45]. Another exploration route would be to model the surface of the crystals as they undergo irradiation.

**Determination of mechanical properties by nanoindentation.** In order to shed light on these predicted mechanical behaviors, we have performed nanoindentation experiments, which give us a concrete foundation to verify the mechano-structure-property

relationship. Characteristic load–displacement ($P - h$) curves and scanning probe microscopy (SPM) images of the nanoindentation impression of crystals **1** and **2** have been obtained (Fig. 6), which are almost identical in nature. The elastic modulus ($E$) and hardness ($H$) have been obtained by standard Oliver–Pharr ($O - P$) method[46], which suggest that both the crystals are soft in nature and thus undergo photocycloaddition upon UV irradiation. The $E$ and $H$ values of crystal **1** ($10.23 \pm 0.78$ GPa and $679.69 \pm 94.87$ MPa) at 1 mN load are found to be marginally higher than that of **2** ($9.59 \pm 0.49$ GPa and $541.05 \pm 44.05$ MPa), but are identical within the range of experimental uncertainty, thus serving as a benchmark for the DFT-predicted values.

## Conclusions

In summary, we have synthesized two metal-organic discrete complexes based on Zn(II) and Cd(II) which undergo photochemical $[2 + 2]$ cycloaddition to generate cyclobutane compounds in a SCSC manner. Of these, the Zn-complex experiences violent mechanical motions (PS effect) such as jumping, breaking, and splitting. Interestingly, even after the PS effect is observed, the crystals still maintain the single crystallinity and structure of a post-PS crystal that can be structurally characterized to show the formation of discrete cyclobutane complex. To the best of our knowledge, this is one of the rare examples of a crystal that demonstrates the PS effect as a metal-organic complex and maintains single crystallinity throughout. On the other hand, the Cd(II) complex does not show such PS effect, providing an ion-mediated mechanism for designing photomechanically active crystalline materials. The Cd(II) crystal also undergoes $[2 + 2]$ photoreaction under UV as well as sunlight to generate a 1D chain of cyclobutane as a crystallized product. DFT calculations reveal that the PS effect can be seen as a mechanism for seeking

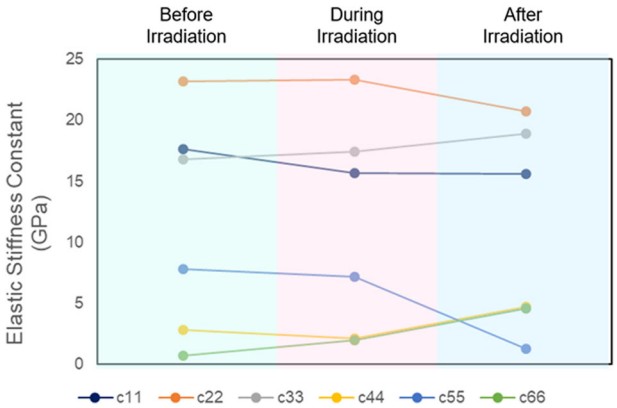

**Fig. 4 Theoretical prediction.** DFT-predicted trends in elastic stiffness constants during UV-induced photosalience.

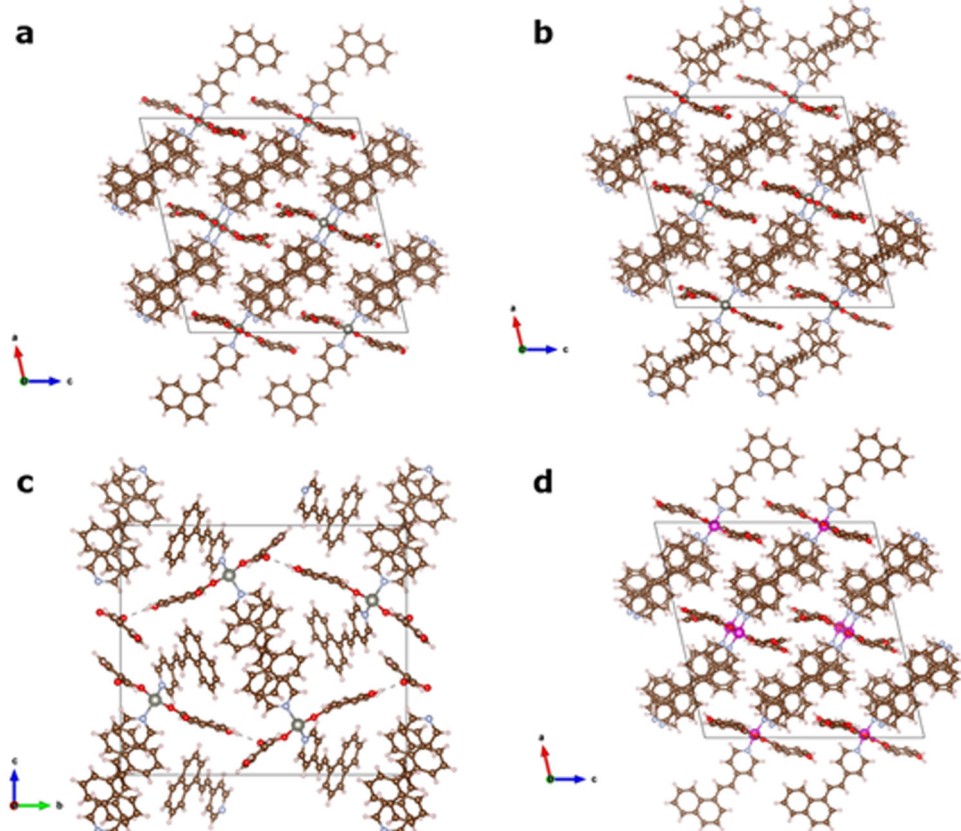

**Fig. 5 Structural prediction.** Crystal structures of (**a**) **1**. **b i1**. **c 1'**. **d 2**. Color code: Zn (silver); Cd (pink); H (white); N (blue); C (brown); O (red).

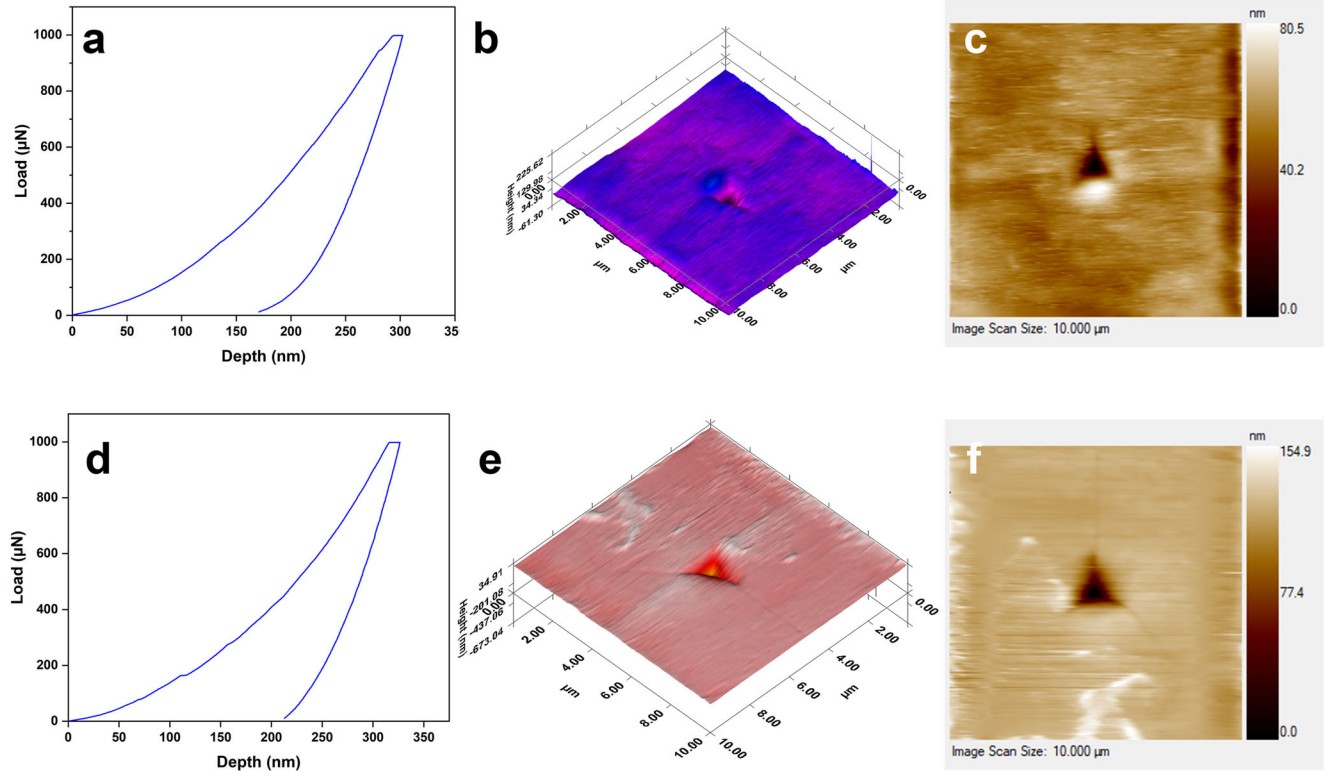

**Fig. 6 Experimental nanoindentation benchmarks of SC mechanical properties. a** The $P-h$ curve of compound **1**. **b** 3D mapped surface of **1**. **c** 2D SPM image of crystal **1**. **d** The $P-h$ curve of compound **2**. **e** 3D mapped surface of **2**. **f** 2D SPM image of crystal **2**.

mechanical stability and higher symmetry, where UV radiation creates a complex network of internal restrictions and relaxations that can only be resolved through mechanical motion. This motion is facilitated by low shear stiffness around two crystallographic axes. The results of the DFT calculations are well validated by nanoindentation studies. Therefore, it can be concluded that the metal ion plays a pivotal role in facilitating the PS effect in these crystals, as well as inherent structural features that give rise to low shear stiffness in the unit cell. The results from the study pave the path for design, development and mechanistic insight into converting the unit cell contractions of microscopic crystals into controlled macroscopic mechanical motions which will expand the field of photoactuating smart materials.

## Materials and methods
All chemicals purchased were reagent grade and were used without further purification. Elemental analysis (carbon, hydrogen and nitrogen) was performed on a Perkin–Elmer 240 C elemental analyzer. $^1$H NMR spectra were recorded on a 400 MHz Bruker Avance 400 FT NMR spectrometer with TMS as an internal reference in DMSO-$d_6$ solution. The solid-state photoluminescence measurements were made using Horiba Fluorolog with a solid-state sample holder. The solid-state photoluminescence measurements were made using Horiba Jobin Yvon Fluoromax-4 Spectrophotometer (Excitation wavelength – 390 nm) with a solid-state sample holder. JEOL JSM-6701F Field Emission Scanning Electron Microscope (FESEM) was used for SEM images. Powder X-ray diffraction (PXRD) data were recorded on a Bruker, D8 ADVANCED X-ray diffractometer with graphite monochromatized Cu Kα radiation (λ = 1.54056 Å) at room temperature (298 K). Photodimerization reaction was carried out using Luzchem photoreactor (8 W UVA lamps) at ~350 nm at room temperature. Crystalline ground powder was packed between the glass slides and irradiated under UV light. Glass slides were flipped at regular intervals of time to maintain uniform exposure of UV radiation. Photosalient effects were studied by irradiating good quality single crystal with UV light of Xe source using a MAX-350 optical photoreactor under a microscope equipped with a high-quality camera for capturing videos.

### Syntheses of compounds
*Synthesis of 1*. A solution of 4-nvp (0.046 g, 0.2 mmol) in MeOH (2 mL) was slowly and carefully layered onto a solution of Zn(NO₃)·6H₂O (0.059 g, 0.2 mmol), in

H₂O (2 mL) using a 2 mL 1 : 1 ( = v/v) solution of MeOH and H₂O followed by layering of 4-ohbz (0.027 g, 0.2 mmol) neutralized with Et₃N (0.021 g, 0.2 mmol) in 2 mL EtOH.The reaction mixture was kept in the dark. The light-yellow color needle shaped crystals of [Zn(4-nvp)₂(4-ohbz)₂] (**1**), were obtained after four days (0.100 g, yield 62%). Elemental analysis (%) calculated for C₄₈H₃₆N₂O₆Zn: C, 71.87; H, 4.52; N, 3.49; found: C 71.6, H 4.2, N 3.7

*Synthesis of i1*. The compound **i1** was synthesized by UV irradiation of **1**: Yellow colored needle single crystals of **1** were irradiated using a UV-lamp (LZC-UVA; Luzchem) centered at ~350 nm wavelength for 5 min to obtain the photodimerized (intermediate) product in almost quantitative yield.

*Synthesis of 1′*. The compound **1′** was synthesized by UV irradiation of **1**: Yellow colored needle single crystals of **1** were irradiated using a UV-lamp (LZC-UVA; Luzchem) centered at ~350 nm wavelength for 30 min to obtain photodimerized product in almost quantitative yield.

*Synthesis of p1*. The block shaped yellow colored thick crystals of [Zn(4-ohbz)₂(4-nvp)₂] (**p1**) were obtained by keeping the reaction mixture of **1** for a month(0.052 g, yield 30%). Elemental analysis (%) calculated for C₄₈H₃₆N₂O₆Zn: C, 71.87; H, 4.52; N, 3.49; found: C 71.2, H 4.1, N 3.8.

*Synthesis of 2*. A solution of 4-nvp (0.046 g, 0.2 mmol) in MeOH (2 mL) was slowly and carefully layered onto a solution of Cd(NO₃)₂·4H₂O (0.062 g, 0.2 mmol), in H₂O (2 mL) using a 2 mL 1 : 1 ( = v/v) solution of MeOH and H₂O followed by layering of 4-ohbz (0.027 g, 0.2 mmol) neutralized with Et₃N (0.021 g, 0.2 mmol) in 2 mL EtOH. The light yellow color block shaped crystals of [Cd(4-nvp)₂(4-ohbz)₂] (**2**),were obtained after seven days (0.110 g, yield 65%). Elemental analysis (%) calculated for C₄₈H₃₆CdN₂O₆: C 67.82, H 4.2, N 3.2; found: C 67.55, H 4.5, N 3.5.

*Synthesis of 2′*. The compound **2′** was synthesized by UV Irradiation of **2**: Yellow colored needle single crystals of **2** were irradiated using a UV-lamp (LZC-UVA; Luzchem) centered at ~350 nm wavelength for 30 min to obtain photo-dimerized product in almost quantitative yield.

**X-ray crystallography**. Single crystals of the all compounds having suitable dimension were used for data collection using a Bruker SMART APEX II diffractometer equipped with graphite-monochromated MoKα radiation (λ = 0.71073 Å). The crystal structure was solved using the SHELXT 2018/6 structure solution program[47]. The collected data (I > 2σ(I)) was integrated by using SAINT[48] program, and the absorption

correction was done by SADABS[49]. Non-hydrogen atoms were refined by the help of anisotropic displacement parameters. All the hydrogen atoms were placed in their geometrically perfect positions and constrained to ride on their parent atoms. Crystallographic data for all compounds are summarized in Supplementary Tables 1–2 and selected bond lengths and bond angles are given in Supplementary Tables 3–7.

**Computational methods**. Elastic constants were calculated using CP2K and were obtained following Vanderbilt's work on the systematic treatment of perturbations[50]. More precisely, the relaxed-ions approach described by Eq. 1 was used to avoid the computation of the pseudo-inverse of the Hessian matrix, and obtain the elastic tensors with a single derivation:

$$C_{ij} = -\frac{\partial \sigma_i}{\partial \eta_j} \qquad (1)$$

A perturbative approach is used to perform the derivation. The system is strained along the 6 Voigt directions with a negative and positive perturbation. The stress tensors are then used to build the aforementioned tensors using finite difference. The unique rotations of the native space group are then applied to the tensors to symmetrize them. The Bulk, Young's and Shear moduli are presented as Voigt-Reuss-Hill averages, and crystal structures were visualized using VESTA.

**Nanoindentation details**. Crystals were mounted using feviquick glue on a stainless-steel round shaped sample holder having smooth surface. The experiment was carried out using a nanoindenter (Hysitron Triboindenter, TI Premier, Minneapolis, USA) with a three-sided pyramidal Berkovich diamond indenter tip of radius 120 nm having an in situ scanning probe microscopy (SPM) facility. Before nanoindentation testing, the tip area function was calculated from a series of indentations on a standard fused quartz sample. The rates of loading and unloading were both 1 mN/s and 5 mN/s with 5 s duration and a 2 s holding period was applied at the maximum indentation depth (Supplementary Figs. 11–12). Several indentations were performed for each crystal and SPM images of the indentation impressions were captured immediately just after unloading. The obtained $P - h$ curves were analyzed using the standard Oliver−Pharr method[46] to extract the required parameters such as elastic modulus ($E$), and hardness ($H$) of the crystals.

## Data availability

The authors declare that all the other data supporting the findings of this study are available within the Article and its Supplementary Information files and from the corresponding author upon request. The X-ray crystallographic coordinates for structures reported in this Article have been deposited at the Cambridge Crystallographic Data Center (CCDC), under deposition number 2214421 (**1**, Supplementary Data 1), 2214422 (**i1**, Supplementary Data 2), 2214423 (**1′**, Supplementary Data 3), 2214424 (**p1**, Supplementary Data 4) 2214425 (**2**, Supplementary Data 5) 2214426 (**2′**, Supplementary Data 6). These data can be obtained free of charge from The Cambridge Crystallographic Data Center via www.ccdc.cam.ac.uk/data_request/cif." CIF files are available as supplementary data files 1–6.

## Code availability

No custom code or mathematical algorithms were required for the central conclusions of this manuscript. Mechanical properties can be calculated using https://www.cp2k.org/ as per the methods section, and https://progs.coudert.name/elate.

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

## Acknowledgements
M.H.M. is thankful to SERB India for a Core Research Grant (Grant No. CRG/2022/001842, dated 19/12/2022) and S.K. thanks CSIR India for awarding a Research Associateship (Sanction No. 09/1157(0005)/2019-EMR-I). R.M. thanks DST-SERB for SRG/2019/001508 and IIT Bhilai for RIG/2003600. S.G. and E.K. would like to acknowledge funding from Science Foundation Ireland under grant number 21/PATH-S/9737. S.G. and P.A.C. acknowledge ongoing support from the Irish Center for High-End Computing (ICHEC) and Science Foundation Ireland award number 12/RC/2275_P2. S.G is funded by the European Union. Views and opinions expressed are however those of the author only and do not necessarily reflect those of the European Union or the European Research Council. Neither the European Union nor the granting authority can be held responsible for them. We also acknowledge Mr. Shamim Ahmad and Prof. C. Malla Reddy, Department of Chemical Sciences, Indian Institute of Science Education and Research (IISER) Kolkata, India for Nanoindentation experiments.

## Author contributions
M.H.M., R. M. and S.G. designed the project. S.K., S.N. and B.D. synthesized the crystals, S.K. and B.D. carried out the X-ray structural data analysis, S.K. and S.N. performed NMR, PXRD measurements, S.K. and A.C. captured videos of photosalient effect. A.C. collected SEM and PL data under the supervision of R.M., E.K. and P.-A.C. performed the computer simulations, M.H.M., R. M. and S.G. directed and supervised the overall project. All authors contributed to writing the manuscript.

## Competing interests
The authors declare no competing interests.
