## [Peer Review File · Communications Chemistry]

Reviewers' comments:

Reviewer #1 (Remarks to the Author):

In this article, the authors reported two iso-structural metal-organic crystals with different photomechanical properties which undergo topochemical [2+2] cycloaddition under UV as well as sunlight. Different photosalient behaviors including jumping, splitting, rolling, and swelling upon UV irradiation were shown in compound 1. DFT calculations were used to understand the atomic-scale mechanisms and mechanical shifts that occur under irradiation. The results are interesting and instructive. Therefore, I recommend the article to be considered for publication after a minor revision via properly addressing the following concerns.

1. In the introduction section, there is a reference to the response of Mexican beans to heat. Please give some examples on the thermomechanical motion of crystals so as to better transition to the photosalient effect of crystals.

2. In the introduction section, when mentioning that SCSC transformations provide exact structural insights into the transformed structures, some more related literatures should be cited: *Nature Comm.*, 2022, 13, 2847; *J. Am. Chem. Soc.*, 2020, 142, 700; *Chem. Comm.*, 2021, 57, 1129.

3. In the article, only the PXRD patterns of 1 and 1' were listed. Please amend the PXRD data of 2 and 2'.

4. The authors mentioned that the photocycloaddition reaction of 2 was completed for only 3 min upon sunlight, whose reaction rate is quite fast. What are the reaction conditions like? What temperature? How long does it take for 2 to fully react under UV light? Please provide a description.

5. From the videos, it was found that small broken crystals were loaded on the surface of crystal 1. Will these small broken crystals affect the overall mechanical motion of crystal?

6. Please check the references, some references are not correctly cited, such as Ref.11. The Ref.11. is not the article from Vittal et al.

7. The references, including page numbers and volumes, did not fit the style of this journal.

8. There are some grammar mistakes and typos, which should be corrected before its acceptance.

Reviewer #2 (Remarks to the Author):

In this work, the authors synthesized two metal-organic complexes based on Zn(II) and Cd(II) for generating cyclobutane compounds undergo photochemical [2+2] cycloaddition. Computational studies revealed that the introduction of different metal ions can regulate the PS effect in these crystals by adjusting the shear stiffness around two crystallographic axes. This work provides some useful data for designing photoactuating smart materials, but it still needs improvements before I can suggest accept. My detailed comments are as follows:

1. Nanoindentation tests are needed to determine the actual moduli of these crystals, so as to verify the rationality of DFT calculations.
2. The parameters in Table 1 need to be defined or explained in more detail for better understanding.
3. The author mentioned "it is an electronic effect that imbues photosalience and a shift in mechanical properties." (line 223) Here, the authors had better provide some references to support this conclusion.

Reviewer #3 (Remarks to the Author):

Development of novel materials with the photosalient effect is interesting research targets because such materials potentially lead to the systems that can convert light energy into mechanical motions. In this submitted manuscript, Khan et al. report on the photosalient effect obtained from crystals of metal complexes with [2+2] photocyclization units (4-nvp: 4-(1-naphthylvinyl)pyridine)). This paper presents interesting results in which the behavior of the isostructural crystals changes upon light irradiation depending on the constituent metal ions (Zn^{2+} and Cd^{2+}). However, there have already been many previous reports on crystalline materials showing the photosalient effect originating from [2+2] photocyclization reactions. Furthermore, the same authors have already reported ("Photomechanical effect in Zn(II) and Cd(II) 1D coordination polymers: photosalient to non-salient behaviour", *Chem. Commun.* 2022, 58, 12102.) similar dramatic changes in the observed photosalient effects depending on the metal ions (Zn^{2+} and Cd^{2+}) for 1D coordination polymer (CP) crystals composed of exactly the same photoreactive units (4-nvp). While the 1D CPs using a dicarboxylic acid ligand (*cis*-1,4-cyclohexanedicarboxylic acid) in the previously published paper (*Chem. Commun.* 2022, 58, 12102.), the submitted paper presents experimental results on discrete metal complexes using monocarboxylic acid (4-hydroxybenzoic acid). In the present paper, there is a new attempt to rationally explain the difference in the photosalient effect based on the mechanical property predicted by DFT calculations. However, the discussion of mechanisms appears to be inadequate. For these reasons, I would not recommend the submitted manuscript for publication in *Communications Chemistry*.

Additional Comments.

- [1] Only the predictions of mechanical properties obtained by DFT calculations are presented, and there is no experimental confirmation that the differences in the mechanical properties exist in the actual crystals as discussed in this paper.
- [2] It would be difficult to discuss differences in photosalient phenomena solely in terms of static mechanical property predictions for thermodynamically stable crystal structures. Dynamic changes, such as changes over time in the non-symmetric structure formed by photochemical reactions starting mainly at the crystal surface, would be more important.

Another approach, such as simulations that include information on time variation, would be necessary.

Our Response to Reviewer's Comments on the Manuscript (COMMSCHEM-22-0569)

Reviewer #1 (Remarks to the Author):

In this article, the authors reported two iso-structural metal-organic crystals with different photomechanical properties which undergo topochemical [2+2] cycloaddition under UV as well as sunlight. Different photosalient behaviors including jumping, splitting, rolling, and swelling upon UV irradiation were shown in compound 1. DFT calculations were used to understand the atomic-scale mechanisms and mechanical shifts that occur under irradiation. The results are interesting and instructive. Therefore, I recommend the article to be considered for publication after a minor revision via properly addressing the following concerns.

Our Response: We are extremely grateful to the Reviewer 1 for appreciation to our manuscript and supporting for publication.

1. In the introduction section, there referred to the response of Mexican beans to heat. Please give some examples on the thermomechanical motion of crystals so as to better transition to the photosalient effect of crystals.

Our Response: As suggested by the Reviewer 1, we have incorporated the relevant references in the introduction part.

2. In the introduction section, when mentioning that SCSC transformations provide exact structural insights into the transformed structures, some more related literatures should be cited: Nature Comm., 2022, 13, 2847; J. Am. Chem. Soc., 2020, 142, 700; Chem. Comm., 2021, 57, 1129.

Our Response: As suggested by the Reviewer 1, we have cited the relevant references in the revised manuscript.

3. In the article, only the PXRD patterns of 1 and 1' were listed. Please amend the PXRD data of 2 and 2'.

Our Response: We have incorporated the PXRD data of 2 and 2' in the revised SI. Thanks.

4. The authors mentioned that the photocycloaddition reaction of 2 was completed for only 3 min upon sunlight, whose reaction rate is quite fast. What are the reaction conditions like? What temperature? How long does it take for 2 to fully react under UV light? Please provide a description.

Our Response: We thank the reviewer 2 for careful observation. The photocycloaddition reaction of **2** was quite faster and it takes only 20 min under UV light at room temperature for full conversion. However, we have taken single crystal and kept for 3 min in scorching sunlight before data collection that gives 100% conversion of **2**.

5. From the videos, it was found that small broken crystals were loaded on the surface of crystal 1. Will these small broken crystals affect the overall mechanical motion of crystal?

Our Response: Thanks for the observation. As per as practical the small crystals on the large crystal do not have any impact on overall mechanical motion of crystal.

6. Please check the references, some references are not correctly cited, such as Ref. 11. The Ref. 11. is not the article from Vittal et al.

Our Response: We are extremely to the reviewer for pointing this out. This has now been corrected.

7. The references, including page numbers and volumes, did not fit the style of this journal.

Our Response: We are sorry for the mistakes. We have now corrected the style of the referenced journals.

8. There are some grammar mistakes and typos, which should be corrected before its acceptance.

Our Response: We have carefully checked the language and format and tried to make it error free as much as possible. Thanks.

Reviewer #2 (Remarks to the Author):

In this work, the authors synthesized two metal-organic complexes based on Zn(II) and Cd(II) for generating cyclobutane compounds undergo photochemical [2+2] cycloaddition. Computational studies revealed that the introduction of different metal ions can regulate the PS effect in these crystals by adjusting the shear stiffness around two crystallographic axes. This work provides some useful data for designing photoactuating smart materials, but it still needs improvements before I can suggest accept. My detailed comments are as follows:

Our Response: We are extremely grateful to the Reviewer 2 for appreciation to our manuscript and supporting for publication.

1. Nanoindentation tests are needed to determine the actual moduli of these crystals, so as to verify the rationality of DFT calculations.

Our Response: Nanoindentation measurements have now been added to the text and correlated to DFT-predicted values. The DFT methodology used has been extensively benchmarked against anisotropic nanoindentation measurements and other experimental characterisation techniques. We have now cited these previous papers and have added our benchmarks and controls to the Supporting Information. The purpose of this work is not to experimentally characterise the mechanical properties of these crystals but to use molecular modelling to rationalise the observed macroscale phenomena. We have added notes on this to the introduction and discussion.

Changes to text

Intro

Outside of our previous work on high-accuracy screening of mechanical properties and phenomena, herein DFT calculations serve to understand the relative internal stress build-up in irradiated photosalient crystal.

Discussion

Our predictions of elastic stiffness tensors have been extensively benchmarked across a number of publications on functional crystals¹⁻³, and experimental mechanical characterisation techniques⁴, allowing for strong confidence in their accuracy and predictability.

Figure S10: Histogram showing a snapshot of the ability of DFT calculations to predict the mechanical properties of materials across multiple crystalline material classes. Note piezoelectric properties are derived from the mechanical stiffness.

2. The parameters in Table 1 need to be defined or explained in more detail for better understanding.

Our Response: Absolutely agree- an extensive explanation has now been added.

Changes to text

The elastic stiffness tensor represents the anisotropic stress strain response of a material⁵, where a tensor component c_{ij} represents the stiffness of a material strained in direction j with an induced stress in direction j ⁶. The numbers 1,2 and 3 correspond to longitudinal responses in Cartesian directions x,y , and z (and also crystallographic axes a,b , and c in cubic or orthorhombic symmetries), and the numbers 4,5, and 6 correspond to shear stress/strain responses along those same respective axes. Our calculations predict the full thirty-six component tensor for each crystal, which is shown in supporting information. For clarity, only the diagonal elements are shown and discussed in the main text i.e. the stress-strain response

along each longitudinal and shear axis. Using the software ELATE the elastic moduli have been derived from this tensor, namely:

The Young's Modulus is by definition a measure of the ability of a crystal to withstand changes in length when under lengthwise tension or compression. It can be thought of as a bulk average of the anisotropic elastic stiffness tensor components, taking into account the dimensions of the system if calculated using the methods of Nye.

The Bulk Modulus of a substance is a measure of how resistant to compression the substance is. It is defined as the ratio of the infinitesimal pressure increase to the resulting relative decrease of the volume i.e. how the volume of the crystal responds to a force acting equally on all faces of the crystal.

The Shear Modulus is a measure of a crystal's resistance or response to shear stress. Similarly to the Young's Modulus the Shear Modulus can be conceptualised of as a bulk average of the shear elastic stiffness tensor contributions, though the mathematics is more complex than this.

These moduli are presented as averages of the widely-used Voigt, Reuss, and Hill approximations^{7,8}, with the standard deviation being across these three approximations.

3. The author mentioned "it is an electronic effect that imbues photosaliency and a shift in mechanical properties." (line 223) Here, the authors had better provide some references to support this conclusion.

Our Response: The reviewer is right, this reads too much like conjecture. We have deleted this line and related our conclusions and future work to the literature with multiple added citations:

Changes to text

The change in mechanical properties during irradiation is a direct result of the aforementioned changes in unit cell parameters. These calculations provide a molecular mechanism for the hypothesis that photosaliency is a "macroscopic manifestation of the stress that can develop in response to the mechanical force created inside a crystal as a result of miniscule structure perturbation"⁹⁻¹¹, Future calculations will expand on this work to look at dynamical stress-strain responses across different ligand-metal combinations, as well as examining volume expansion effects.^{12,13}

Reviewer #3 (Remarks to the Author):

Development of novel materials with the photosalient effect is interesting research targets because such materials potentially lead to the systems that can convert light energy into mechanical motions. In this submitted manuscript, Khan et al. report on the photosalient effect obtained from crystals of metal complexes with [2+2] photocyclization units (4-nvp: 4-(1-naphthylvinyl)pyridine}). This paper presents interesting results in which the behavior of the isostructural crystals changes upon light irradiation depending on the constituent metal ions (Zn^{2+} and Cd^{2+}). However, there have already been many previous reports on crystalline materials showing the photosalient effect originating from [2+2] photocyclization reactions. Furthermore, the same authors have already reported ("Photomechanical effect in Zn(II) and Cd(II) 1D coordination polymers: photosalient to non-salient behaviour", Chem. Commun. 2022, 58, 12102.) similar dramatic changes in the observed photosalient effects depending on the metal ions (Zn^{2+} and Cd^{2+}) for 1D coordination polymer (CP) crystals composed of exactly the same photoreactive units (4-nvp). While the 1D CPs using a dicarboxylic acid ligand (cis-1,4-cyclohexanedicarboxylic acid) in the previously published paper (Chem. Commun. 2022, 58, 12102.), the submitted paper presents experimental results on discrete metal complexes using monocarboxylic acid (4-hydroxybenzoic acid). In the present paper, there is a new attempt to rationally explain the difference in the photosalient effect based on the mechanical property predicted by DFT calculations. However, the discussion of mechanisms appears to be inadequate. For these reasons, I would not recommend the submitted manuscript for publication in Communications Chemistry.

Our Response: We respect the decision made by the learned Reviewer 3. However, we have revised manuscript incorporating further commentary on the mechanisms of photosalience and how this correlates to the supramolecular packing of these systems, including linking our observations to the latest work in the field, i.e. we have thoroughly expanded the mechanistic discussion in the paper.

Changes to text

From a supramolecular packing perspective, photosalience facilitates a significant rearrangement of the 4-ohbz chains. The increased volume and change in symmetry observed in crystal I' allows these molecular moieties to relax and stretch out (Figure 5c), preferring this to

the constrained ‘wrinkled’ arrangement of crystal 1 and 1I. These calculations provide a molecular mechanism for the hypothesis that photosaliency is a “macroscopic manifestation of the stress that can develop in response to the mechanical force created inside a crystal as a result of miniscule structure perturbation”⁹⁻¹¹

Besides, in order to shed light to the mechanical properties, we have performed nanoindentation experiments, which give us a concrete idea to establish the mechano-structure-property relationship. The nanoindentation results are well corroborated with the DFT-predicted values. Hope the results described in the revised manuscript will have a significant impact among researchers working in the fields of Supramolecular Chemistry, Photochemistry, Crystal Engineering and Materials Chemistry.

Changes to text

Determination of mechanical properties by nanoindentation

In order to shed light to the mechanical properties, we have performed nanoindentation experiments, which give us a concrete idea to establish the mechano-structure-property relationship. Characteristic load–displacement (P–h) curves and scanning probe microscopy (SPM) images of the nanoindentation impression of crystals 1 and 2 have been obtained (Fig. 6), which are almost identical in nature. The elastic modulus (E) and hardness (H) have been obtained by standard Oliver–Pharr (O–P) method,¹⁴ which suggest that both the crystals are soft in nature and thus undergo photocycloaddition upon UV irradiation. The E and H values of crystal 1 (10.23 ± 0.78 GPa and 679.69 ± 94.87 MPa at 1 mN load) are found to be marginally higher than that of 2 (9.59 ± 0.49 GPa and 541.05 ± 44.05 MPa), which signifies higher resistance towards flexibility of crystal 1 leading to photosalient property. However, as expected crystal 2 having more flexibility does not exhibit photosaliency. Therefore, nanoindentation results are well corroborated with the DFT-predicted values.

Additional Comments.

[1] Only the predictions of mechanical properties obtained by DFT calculations are presented, and there is no experimental confirmation that the differences in the mechanical properties exist in the actual crystals as discussed in this paper.

Our Response: Nanoindentation measurements have now been added to the text and correlated to DFT-predicted values. The DFT methodology used has been extensively benchmarked against anisotropic nanoindentation measurements and other experimental characterisation techniques. We have now cited these previous papers and have added our benchmarks and controls to the Supporting Information. The purpose of this work is not to experimentally characterise the mechanical properties of these crystals but to use molecular modelling to rationalise the observed macroscale phenomena. We have added notes on this to the introduction and discussion.

[2] It would be difficult to discuss differences in photosolient phenomena solely in terms of static mechanical property predictions for thermodynamically stable crystal structures. Dynamic changes, such as changes over time in the non-symmetric structure formed by photochemical reactions starting mainly at the crystal surface, would be more important. Another approach, such as simulations that include information on time variation, would be necessary.

Our Response: A very interesting point- we think one of the strengths in this study is that we obtained a high-resolution XRD structure of the crystals as they are undergoing photosolience, which means that we can use this series of technically static data points to show how this phenomenon evolves over time- while something like molecular dynamics would give us 500ns of dynamic data, we wouldn't learn as much, particularly from an anisotropic standpoint, as we do from comparing these static time stamps. Computational studies of surface irradiation would be fascinating, we have suggested this as future work.

Changes to text

By obtaining a high-resolution XRD structure of the crystals as they are undergoing photosolience, we can use DFT to obtain dynamic information about how this phenomenon evolves over time, even with thermodynamically stable crystal structures.

Future calculations will expand on this work to look at dynamical stress-strain responses across different ligand-metal combinations, as well as examining volume expansion effects.^{12,13} Another exploration route would be to model the surface of the crystals as they undergo irradiation.

Added references

- 1 Guerin, S., Tofail, S. A. & Thompson, D. Longitudinal piezoelectricity in natural calcite materials: Preliminary studies. *IEEE Transactions on Dielectrics and Electrical Insulation* **25**, 803-807 (2018).
- 2 Hasija, A. *et al.* Tracing shape memory effect and elastic bending in a conformationally flexible organic salt. *Journal of Materials Chemistry C* **10**, 4257-4267 (2022).
- 3 O'Donnell, J. *et al.* Atomistic-Benchmarking towards a protocol development for rapid quantitative metrology of piezoelectric biomolecular materials. *Applied Materials Today* **21**, 100818 (2020).
- 4 Kiely, E., Zwane, R., Fox, R., Reilly, A. M. & Guerin, S. Density functional theory predictions of the mechanical properties of crystalline materials. *CrystEngComm* **23**, 5697-5710 (2021).
- 5 Pelleg, J. *Mechanical properties of materials*. Vol. 190 (Springer, 2013).
- 6 Nye, J. F. *Physical properties of crystals: their representation by tensors and matrices*. (Oxford university press, 1985).
- 7 Chung, D. & Buessem, W. The Voigt-Reuss-Hill (VRH) approximation and the elastic moduli of polycrystalline ZnO, TiO₂ (Rutile), and α -Al₂O₃. *Journal of applied physics* **39**, 2777-2782 (1968).
- 8 Hill, R. The elastic behaviour of a crystalline aggregate. *Proceedings of the Physical Society. Section A* **65**, 349 (1952).
- 9 Naumov, P., Sahoo, S. C., Zakharov, B. A. & Boldyreva, E. V. Dynamic single crystals: kinematic analysis of photoinduced crystal jumping (the photosalient effect). *Angewandte Chemie* **125**, 10174-10179 (2013).
- 10 Rath, B. B. & Vittal, J. J. Single-crystal-to-single-crystal [2+ 2] photocycloaddition reaction in a photosalient one-Dimensional coordination polymer of Pb (II). *Journal of the American Chemical Society* **142**, 20117-20123 (2020).
- 11 Yadava, K. *et al.* Extraordinary anisotropic thermal expansion in photosalient crystals. *IUCrJ* **7**, 83-89 (2020).
- 12 Hean, D., Alde, L. G. & Wolf, M. O. Photosalient and thermosalient crystalline hemithioindigo-anthracene based isomeric photoswitches. *Journal of Materials Chemistry C* **9**, 6789-6795 (2021).
- 13 Medishetty, R., Sahoo, S. C., Mulijanto, C. E., Naumov, P. & Vittal, J. J. Photosalient behavior of photoreactive crystals. *Chemistry of Materials* **27**, 1821-1829 (2015).

- 14 Oliver, W. C. & M. Pharr, An improved technique for determining hardness and elastic modulus using load and displacement sensing indentation experiments. *J. Mater. Res.* **7**, 1564 (1992).

REVIEWERS' COMMENTS:

Reviewer #1 (Remarks to the Author):

The authors properly responded to the comments raised by the three reviewers and made suitable corrections in the revised MS. I believe the version of this manuscript could be accepted and published in this journal with no further revision.

Reviewer #2 (Remarks to the Author):

The authors have conducted a satisfactory revision, I appreciate the efforts they made. So, I support publication of this version without change.

Reviewer #3 (Remarks to the Author):

The authors have appropriately revised the manuscript, taking into account the reviewers' comments. In particular, I would appreciate the authors' effort for performing the nanoindentation experiments. Through this paper, the authors are successfully demonstrating that molecular modeling based on computational methods can be useful in predicting and explaining macroscopic properties and is expected to attract the attention of many researchers in this related field. Therefore, I would recommend the revised manuscript to be considered for publication in Communications Chemistry.

Our Response to Reviewer's Comments on the Manuscript (COMMSCHEM-22-0569A-Z)

REVIEWERS' COMMENTS:

Reviewer #1 (Remarks to the Author):

The authors properly responded to the comments raised by the three reviewers and made suitable corrections in the revised MS. I believe the version of this manuscript could be accepted and published in this journal with no further revision.

Our Response: We are extremely grateful to the Reviewer 1 for appreciation to our manuscript and supporting for publication.

Reviewer #2 (Remarks to the Author):

The authors have conducted a satisfactory revision, I appreciate the efforts they made. So, I support publication of this version without change.

Our Response: We are extremely grateful to the Reviewer 2 for appreciation to our manuscript and supporting for publication.

Reviewer #3 (Remarks to the Author):

The authors have appropriately revised the manuscript, taking into account the reviewers' comments. In particular, I would appreciate the authors' effort for performing the nanoindentation experiments. Through this paper, the authors are successfully demonstrating that molecular modeling based on computational methods can be useful in predicting and explaining macroscopic properties and is expected to attract the attention of many researchers in this related field. Therefore, I would recommend the revised manuscript to be considered for publication in Communications Chemistry.

Our Response: We are extremely grateful to the Reviewer 3 for appreciation to our manuscript and supporting for publication.